# SWNet: Surgical Workflow Recognition with Deep Convolutional Network

**Bokai Zhang**                                                          BZHANG29@ITS.JNJ.COM
**Amer Ghanem**                                                         AGHANEM1@ITS.JNJ.COM
**Alexander Simes**                                                  ALEX.SIMES29@GMAIL.COM
**Henry Choi**                                                            HCHOI51@ITS.JNJ.COM
**Andrew Yoo**                                                        BANYANFIG@GMAIL.COM
**Andrew Min**                                                             AMIN2@ITS.JNJ.COM
*C-SATS, Inc., Johnson & Johnson, Seattle, WA, USA*

## Abstract

Surgical workflow recognition has been playing an essential role in computer-assisted interventional systems for modern operating rooms. In this paper, we present a computer vision-based method named SWNet that focuses on utilizing spatial information and temporal information from the surgical video to achieve surgical workflow recognition. As the first step, we utilize Interaction-Preserved Channel-Separated Convolutional Network (IP-CSN) to extract features that contain spatial information and local temporal information from the surgical video through segments. Secondly, we train a Multi-Stage Temporal Convolutional Network (MS-TCN) with those extracted features to capture global temporal information from the full surgical video. Finally, by utilizing Prior Knowledge Noise Filtering (PKNF), prediction noise from the output of MS-TCN is filtered. We evaluate SWNet for Sleeve Gastrectomy surgical workflow recognition. SWNet achieves 90% frame-level accuracy and reaches a weighted Jaccard Score of 0.8256. This demonstrates that SWNet has considerable potential to solve the surgical workflow recognition problem.

**Keywords:** surgical workflow recognition, computer-assisted interventional systems, IP-CSN, MS-TCN

## 1. Introduction

Video-based automatic surgical workflow recognition is one of the key technologies to build computer-assisted interventional systems for modern operating rooms. Such systems can enhance coordination among OR teams and improve surgical safety. For offline surgical workflow recognition, it provides a tool to automate the indexing of surgical video databases as well as provides support in Video-Based Assessment (VBA) systems to surgeons for life-long learning (Feldman et al., 2020).

Early studies (Twinanda et al., 2016; Kitaguchi et al., 2020) focused on utilizing image classification networks to capture spatial information from surgical videos on a frame by frame basis to achieve surgical workflow recognition. With the rise of the Recurrent Neural Network, researchers have proposed using 2D Convolutional Neural Network and Recurrent Neural Network together to capture both spatial and temporal information from the surgical video through segments (Jin et al., 2017; Zisimopoulos et al., 2018; Chen et al., 2018; Yengera et al., 2018; Funke et al., 2018; Mondal et al., 2019; Jin et al., 2020; Nakawala et al., 2019; Yi

and Jiang, 2019). In a recent study, Czempiel et al. (2020) utilize ResNet50 (He et al., 2016) as the 2D Convolutional Neural Network to extract visual features frame by frame from the surgical video to capture spatial information. They also utilize a 2-stage causal Temporal Convolutional Network to capture global temporal information from the extracted features for surgical workflow recognition.

In this paper, instead of utilizing ResNet to capture spatial features frame by frame, we implement a deep 3D Convolutional Neural Network named Interaction-Preserved Channel-Separated Convolutional Network (IP-CSN) (Tran et al., 2019) to capture spatial and local temporal features by video segment. We utilize a Multi-Stage Temporal Convolutional Network (MS-TCN) (Farha and Gall, 2019) to capture global temporal information from the video. For offline surgical workflow recognition, we utilize the Prior Knowledge Noise Filtering (PKNF) algorithm to filter the prediction noise from MS-TCN output. We name this IPCSN-MSTCN-PKNF workflow SWNet.

## 2. Methods

### 2.1. Datasets

We test our method of surgical workflow recognition on Sleeve Gastrectomy videos. Sleeve Gastrectomy is used to assist patients with losing excess weight. It can reduce the risk of potentially life-threatening weight-related health problems including type 2 diabetes, high blood pressure, sleep apnea, and more. Our medical experts reviewed the literature on Sleeve Gastrectomy surgical workflow (Iannelli et al., 2008; Daskalakis and Weiner, 2009; Van Rutte et al., 2017; van Ramshorst et al., 2017; Kaijser et al., 2018) and split Sleeve Gastrectomy into 8 surgical phases: "Exploration phase", "Ligation of short gastric vessels phase", "Gastric transection phase", "Bougie phase", "Suturing of omentum to stomach phase", "Liver retraction phase", "Hiatal hernia repair phase", and "Gastric band removal phase". The time interval between surgical phases was named as "Not a surgical phase".

We collected 461 Sleeve Gastrectomy surgical videos from 14 institutions for our dataset. The dataset was split randomly. 317 videos were used for the training dataset. 82 videos were used for the validation dataset. 62 videos were used for the test dataset. Each video is annotated with the above-mentioned set of phases. More details about the datasets are shown in Appendix A.

### 2.2. SWNet for offline surgical workflow recognition

The overview of our SWNet is illustrated in Figure 1. During inference, we divide the video into short video segments and utilize IP-CSN to extract features for each video segment. Each feature can be considered as a summary of the video segment. We concatenate the extracted features to get the full video features and utilize MS-TCN to achieve initial surgical phase segmentation for the full surgical video. We apply the Prior Knowledge Noise Filtering algorithm to the initial surgical phase segmentation results to get the final prediction results for the full video. With SWNet, we are able to capture spatial and local temporal information in short video segments with IP-CSN as well as capture global temporal information in the full video with MS-TCN.

Next, we work on building SWNet for the offline surgical workflow recognition pipeline. We first conduct transfer learning on our dataset with IP-CSN. Then, we utilize IP-CSN to extract features for our dataset. After that, we train the MS-TCN with the extracted features. Finally, we utilize the Prior Knowledge Noise Filtering algorithm to filter the prediction noise from MS-TCN output.

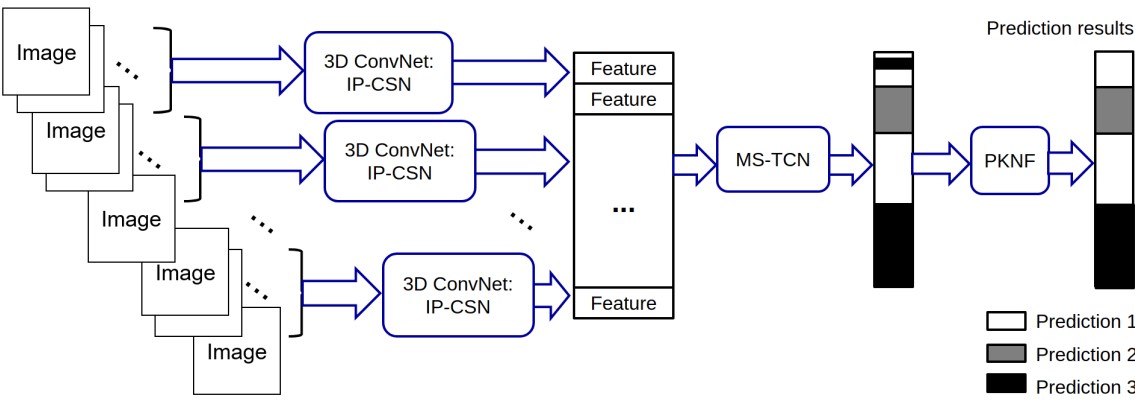

Figure 1: The overview of SWNet

### 2.2.1. IP-CSN as feature extraction backbone

In recent research, 3D ConvNet is used to capture spatial and temporal information in video segments. Carreira and Zisserman (2017) proposed to inflate 2D CNN along the temporal dimension to obtain Inflated 3D ConvNet (I3D). With RGB stream and optical flow stream as the input streams, a two-stream I3D solution is designed. To lower the computational cost and improve accuracy, R(2+1)D (Tran et al., 2018) is designed to factor 3D convolution in space and time while Channel-Separated Convolutional Network (CSN) (Tran et al., 2019) is designed to factor 3D convolution by separating channel interaction and spatiotemporal interaction.

From recent studies (Tran et al., 2019; Ghadiyaram et al., 2019), CSN outperforms two-stream I3D and R(2+1)D on Kinetics-400 dataset (Kay et al., 2017). With large-scale weakly-supervised pre-training on IG-65M dataset (Ghadiyaram et al., 2019), CSN model performs even better. From the computation standpoint, CSN only needs the RGB stream as input while the optical flow stream in two-stream I3D needs expensive computation. Inspired by this, we adopt CSN, specifically, Interaction-Preserved Channel-Separated Convolutional Network for our problem. Appendix B shows the design of IP-CSN bottleneck block.

A large amount of video data is needed for training a 3D ConvNet from scratch, so we conduct transfer learning instead. The initial weights for IP-CSN152 are publicly available (Tran et al., 2019). We utilize the initial weights pretrained on IG-65M and Kinetics-400 for our work. We annotated each of our surgical videos with nine class labels, including eight surgical phase labels and one not surgical phase label. The start time and the end

time for each label are annotated. To fine-tune IP-CSN on our dataset, during each training epoch, five 19.2s video segments are randomly selected inside each annotation segment for each video. 32 frames are sampled with constant intervals as one training sample from each video segment.

### 2.2.2. MS-TCN for surgical phase segmentation

To capture global temporal information from the video, instead of utilizing a 2-stage causal Temporal Convolutional Network proposed in Czempiel et al. (2020), we utilize a 4-stage acausal Temporal Convolutional Network proposed in Farha and Gall (2019). Given the input $X = \{x_1, x_2, \ldots, x_t\}$, MS-TCN predicts the output $P = \{P_1, P_2, \ldots, P_t\}$ where $t$ is the current time step, $1 \leq t <= T$, $T$ is the number of total time steps, $x_t$ is the feature input at time step $t$, $P_t$ is output prediction for the current time step.

The overview of MS-TCN is illustrated in Figure 5 in Appendix D. For the classification loss in MS-TCN, the cross-entropy loss is calculated by

$$L_{cls} = \frac{1}{T} \sum_t -\log(p_{t,c})$$
(1)

where $p_{t,c}$ is the predicted probability at class $c$ at time step $t$. For the smooth loss to reduce over-segmentation, the truncated mean squared error is calculated over the frame-wise log-probabilities by

$$L_{T-MSE} = \begin{cases} \frac{1}{TC} \sum_{t,c} |\log(p_{t,c}) - \log(p_{t-1,c})|^2 & |\log(p_{t,c}) - \log(p_{t-1,c})| \leq \tau \\ \frac{1}{TC} \sum_{t,c} \tau^2 & \text{otherwise} \end{cases}$$
(2)

where $C$ is the total number of classes, $\tau$ is the threshold value. The final loss function sums the losses over all stages which can be calculated by

$$L_{final} = \sum_S (L_{cls} + \lambda L_{T-MSE})$$
(3)

where $S$ is the total stage number for MS-TCN, $\lambda$ is a weighted parameter.

### 2.2.3. Prior Knowledge Noise Filtering

In the surgical videos we gathered, it is observed that surgeons can idle or pull out surgical tools in the middle of a surgical phase. For those video segments, the deep learning model sometimes fails to predict accurately. As a result, we develop a filtering algorithm to filter the wrong predictions.

We develop Prior Knowledge Noise Filtering (PKNF) for offline surgical workflow recognition in consideration of three aspects: surgical phase order, surgical phase incidence, and surgical phase time. From the surgical phase order aspect, we notice several surgical phases follow a specific order. When a prediction from MS-TCN does not follow the specific phase order it should, we correct the prediction by selecting a label that the model has the highest confidence in from the possible labels according to phase order. From the surgical phase time aspect, we run statistical analyses on our annotation to get the minimum phase time $T$. $T = \{T_1, T_2, \ldots, T_N\}$ where $N$ is the total number of the surgical phases. We check that

the prediction segments share the same prediction labels from MS-TCN first. For adjacent prediction segments that share the same prediction labels, we connect them if the time interval between the prediction segments is shorter than the connection threshold we set for that surgical phase. The connection threshold is set according to the minimum phase time $T$. In this work, we set the connection threshold to be 40% of the minimum phase time $T$. After adjacent prediction segments are connected correctly, surgical phase time can be calculated for each surgical phase prediction segment. We correct prediction segments that are too short to be a surgical phase. From the surgical phase incidence aspect, we notice some surgical phases normally only happen less than a fixed incidence number when we run statistical analyses on our annotation. If multiple segments of the same phase show up in the prediction and pass the phase incidence threshold value we set for that surgical phase, we select segments according to the ranking of the model's confidence.

### 2.3. Online surgical workflow recognition

As shown in Figure 1, the final step in SWNet is PKNF. It is an inference algorithm designed specifically for offline surgical workflow recognition. If we take out the PKNF step in SWNet, we can utilize IPCSN-MSTCN for online surgical workflow recognition. During online inference, spatial and local temporal features extracted by IP-CSN are saved by video segment. So at time step $t$, we can read in all features before time step $t$ together with the feature extracted at time step $t$ to build feature set $F = \{f_1, f_2, \ldots, f_t\}$, we send the feature set $F$ to MS-TCN to get prediction output $P = \{P_1, P_2, \ldots, P_t\}$, where $P_t$ is the online prediction result at time step $t$.

## 3. Experimental Results

We utilize IP-CSN152, MS-TCN, and PKNF to build IPCSN-MSTCN-PKNF workflow and refer to it as SWNet. For fine-tuning IP-CSN152, SGD optimizer with an initial learning rate of $2e^{-4}$ is used. We reduce the learning rate by a factor of 0.2 if validation accuracy does not improve in the last 10 epochs. The weight decay is set to be $1e^{-7}$. Random crop is used for data augmentation. For each training sample sequence of frames, we resize the frames according to the smaller side of the frames to 182 pixels and randomly crop 160* 160 patches from them. For data augmentation, random rotation is applied to 10% of the training samples. Random flipping is applied to 10% of the training samples. After training the IP-CSN152, we remove the final layer of IP-CSN152 and use it to extract 2048-dimensional feature vectors. We utilize acausal Temporal Convolutional Networks for MS-TCN. The total number of stages is set to be 4. The total number of dilated convolution layers at each stage is set to be 10. The number of feature maps is set to be 64. During the training of MS-TCN, Adam optimizer is used with a learning rate of $5e^{-4}$.

To quantify the importance of utilizing local temporal information from the feature extraction backbone, we replace IP-CSN152 from SWNet with a 2D ConvNet named EfficientNet-B5 (Tan and Le, 2019) to build EfficientNet-MSTCN-PKNF workflow. We utilize EfficientNet-B5 to capture spatial information only as the feature extraction backbone. For fine-tuning EfficientNet-B5, SGD optimizer with an initial learning rate of $1e^{-4}$ is used. We reduce the learning rate by a factor of 0.2 if validation accuracy does not improve in the last 10 epochs. For each training sample frame, we resize the frame according to the smaller side

of the frame to 510 pixels and randomly crop 456*456 patch from it. Random rotation and random flipping are also used during the training of EfficientNet-B5.

To quantify the importance of utilizing MS-TCN as the video action segmentation network, we replace MS-TCN from SWNet with a two-layer LSTM to build the IPCSN-LSTM-PKNF workflow. The hidden unit size for LSTM is set to be 128. The dropout rate is set to be 0.5. The learning rate is set to be 0.005.

### 3.1. Results for offline surgical workflow recognition

We evaluate our methods against ResNetLSTM (Jin et al., 2017) as well as TeCNO (Czempiel et al., 2020). The overall experimental results conducted on our test dataset are shown in Table 1. Our EfficientNet-MSTCN-PKNF outperforms both ResNetLSTM and TeCNO. By utilizing PKNF, SWNet outperforms IPCSN-MSTCN by 1.16% and 0.0186 in terms of the overall accuracy and the weighted Jaccard Score. IPCSN-LSTM-PKNF outperforms IPCSN-LSTM by 1.65% and 0.0239 in terms of the overall accuracy and the weighted Jaccard Score. The above results show that utilizing PKNF in the workflow can reduce noise and improve prediction results. SWNet outperforms EfficientNet-MSTCN-PKNF by 1.76% and 0.0261 in terms of the overall accuracy and the weighted Jaccard Score. This shows that IP-CSN is a better feature extraction backbone compares to EfficientNet. SWNet outperforms IPCSN-LSTM-PKNF by 3.24% and 0.0512 in terms of the overall accuracy and the weighted Jaccard Score. This shows that MS-TCN is a better video action segmentation network compares to LSTM. The mean accuracy, the standard deviation of the accuracy, the mean weighted Jaccard Score, and the standard deviation of the weighted Jaccard Score are shown in Table 4 in Appendix C. SWNet outperforms all other methods in terms of the mean accuracy and the mean weighted Jaccard Score.

To further compare the performance between ResNetLSTM, TeCNO, EfficientNet-MSTCN-PKNF, IPCSN-LSTM-PKNF, and SWNet, we calculate Precision, Recall, and F1-Score for each surgical phase in Table 5 in Appendix C. Except for the "Exploration phase", the F1-Score for SWNet outperforms the F1-Score for other networks. SWNet performs well in most surgical phases. From Table 3 in Appendix A, lack of training data might be the reason why SWNet does not perform well in some surgical phases.

As shown in Figure 2, we visualize the predictions results from ResNetLSTM, TeCNO, EfficientNet-MSTCN, EfficientNet-MSTCN-PKNF, IPCSN-LSTM, IPCSN-LSTM-PKNF, IPCSN-MSTCN, and IPCSN-MSTCN-PKNF (SWNet) from 4 test videos for offline surgical workflow recognition. It is clear that SWNet can locate the surgical phase more accurately and identify phase transactions better.

### 3.2. Results for online surgical workflow recognition

We also evaluate IPCSN-MSTCN for online surgical workflow recognition. Instead of focusing on utilizing weighted cross-entropy loss in Czempiel et al. (2020), we investigate the effect of applying smoothing loss in Farha and Gall (2019).

We evaluate our methods against ResNetLSTM (Jin et al., 2017) as well as TeCNO (Czempiel et al., 2020). As shown in Table 2, the performance for TeCNO and IPCSN-MSTCN are similar from the overall accuracy and the weighted Jaccard Score aspects. Models achieving similar accuracy may have large differences, as visualized in Figure 3.

Table 1: Overall accuracy and Jaccard score for offline surgical workflow recognition

| Method | Accuracy | Weighted Jaccard Score |
|---|---|---|
| ResNetLSTM | 0.8235 | 0.7141 |
| TeCNO | 0.8659 | 0.7668 |
| EfficientNet-MSTCN | 0.8818 | 0.7928 |
| EfficientNet-MSTCN-PKNF | 0.8861 | 0.7995 |
| IPCSN-LSTM | 0.8548 | 0.7505 |
| IPCSN-LSTM-PKNF | 0.8713 | 0.7744 |
| IPCSN-MSTCN | 0.8921 | 0.8070 |
| IPCSN-MSTCN-PKNF (SWNet) | 0.9037 | 0.8256 |

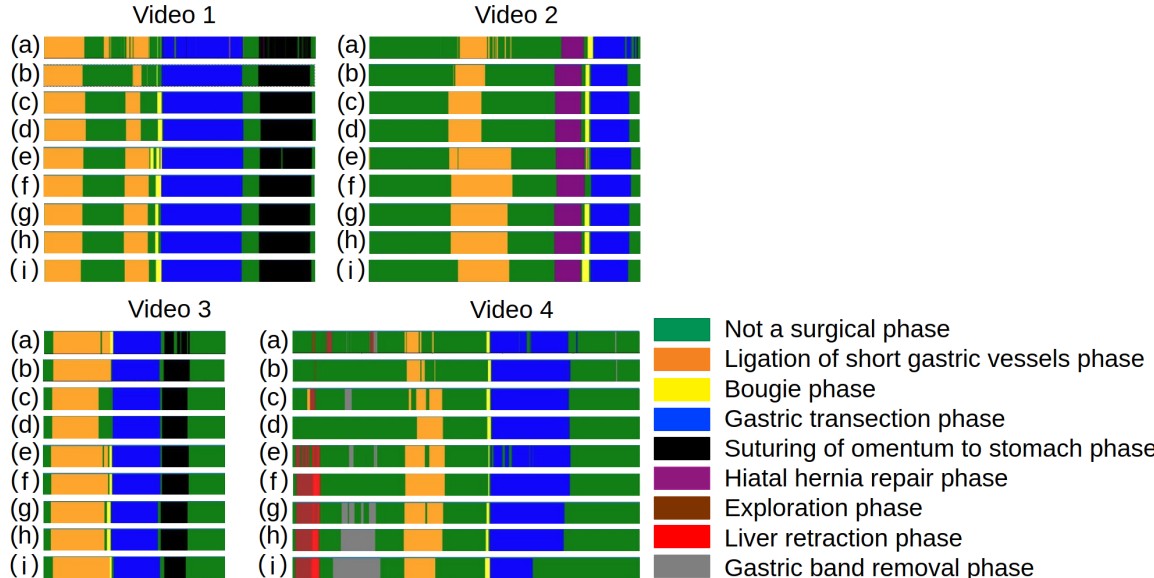

Figure 2: Color-coded ribbon illustration for offline recognition results: (a) ResNetL-STM prediction results (b) TeCNO prediction results (c) EfficientNet-MSTCN model output (d) EfficientNet-MSTCN-PKNF prediction results (e) IPCSN-LSTM model output (f) IPCSN-LSTM-PKNF prediction results (g) IPCSN-MSTCN model output (h) SWNet prediction results (i) Ground Truth

These frame-wise metrics are not suitable to evaluate over-segmentation errors. In order to evaluate out-of-order predictions and over-segmentation errors, segmental metrics (Lea et al., 2016, 2017; Farha and Gall, 2019) are utilized. We calculate the segmental edit distance score, and the segmental F1 score at overlapping thresholds 10%, 25%, and 50% as shown in Appendix E. The overlapping threshold is determined based on the intersection over union (IoU) ratio. After applying smooth loss, the segmental edit distance score and the segmental F1 score for IPCSN-MSTCN improve a lot. Comparing with ResNetLSTM and TeCNO, IPCSN-MSTCN trained with smoothing loss can provide smoother predictions.

As shown in Figure 3, we visualize the predictions results from ResNetLSTM and TeCNO together with the output of the predictions from IPCSN-MSTCN trained with different loss functions. It is clear that applying smooth loss can alleviate over-segmentation errors for online surgical workflow recognition.

Table 2: Overall accuracy, segmental edit distance and segmental F1 for online surgical workflow recognition

| Method | Accuracy | Jaccard | Edit | F1@10 | F1@25 | F1@50 |
|---|---|---|---|---|---|---|
| ResNetLSTM | 0.8130 | 0.6997 | 22.2775 | 23.2044 | 20.6931 | 15.7710 |
| TeCNO | 0.8451 | 0.7331 | 42.5531 | 46.7005 | 43.8578 | 35.7360 |
| IPCSN-MSTCN($L_{cls}$) | 0.8425 | 0.7326 | 49.5681 | 49.6224 | 44.8759 | 33.6570 |
| IPCSN-MSTCN($L_{cls} + \lambda L_{T-MSE}$) | 0.8466 | 0.7367 | 56.5213 | 56.1170 | 52.9255 | 41.4894 |

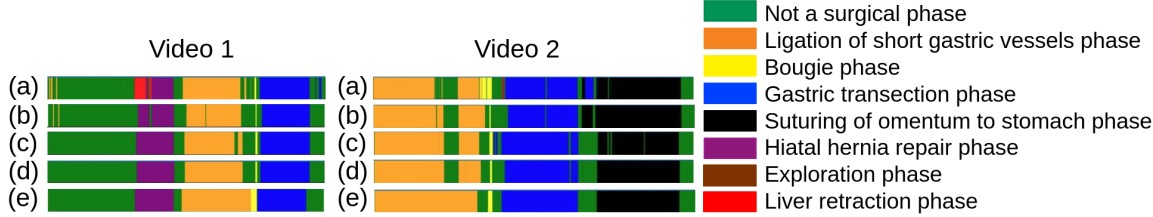

Figure 3: Color-coded ribbon illustration for online recognition results: (a) ResNetLSTM prediction results (b) TeCNO prediction results (c) Predictions from IPCSN-MSTCN trained with $L_{cls}$ (d) Predictions from IPCSN-MSTCN trained with $L_{cls} + \lambda L_{T-MSE}$ (e) Ground Truth

## 4. Conclusion

In this paper, we designed SWNet for surgical workflow recognition with IP-CSN, MS-TCN, and PKNF. We show that utilizing IP-CSN with RGB stream outperforms EfficientNet as the feature extraction backbone. We show that PKNF can improve prediction results for offline surgical workflow recognition as well as applying smooth loss can reduce over-segmentation errors for online surgical workflow recognition. For future work, we want to investigate prediction filtering algorithms like PKI (Jin et al., 2017) for online surgical workflow recognition. We want to conduct a deeper analysis for the prediction errors and conduct more experiments on other procedures for surgical workflow recognition.

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

## Appendix A. Details about the datasets

We calculate the minutes of video data we have for the dataset. As shown in Table 3, we have a limited amount of data for several surgical phases like: "Exploration phase", "Bougie phase", "Liver retraction phase", and "Gastric band removal phase". There are two reasons that cause this data imbalance problem. One reason is the operation time for different surgical phases varies from one another. Another reason is that surgical phases like "liver retraction" are optional during the surgery. Video segments labeled as "Not a surgical phase" are usually surgical phase transaction segments, undefined surgical phase segments, out-of-body segments, idle segments, and so on.

Table 3: Training, validation and test datasets (minutes of video)

| Phase Name | Training Data | Validation Data | Testing Data |
| --- | --- | --- | --- |
| Not a surgical phase | 5729.91 | 1460.91 | 1202.01 |
| Ligation of short gastric vessels phase | 4247.63 | 1082.03 | 828.13 |
| Gastric transection phase | 3988.37 | 953.85 | 690.50 |
| Bougie phase | 305.08 | 64.35 | 50.62 |
| Suturing of omentum to stomach phase | 2562.70 | 807.70 | 397.62 |
| Exploration phase | 181.83 | 38.33 | 27.22 |
| Liver retraction phase | 65.48 | 25.97 | 6.88 |
| Hiatal hernia repair phase | 448.95 | 72.38 | 102.63 |
| Gastric band removal phase | 52.63 | 42.32 | 31.03 |

## Appendix B. IP-CSN block example

CSN (Tran et al., 2019) is defined as 3D CNNs in which all convolutional layers (except for conv1) are either $1 \times 1 \times 1$ conventional convolutions or $k \times k \times k$ depthwise convolutions. $1 \times 1 \times 1$ conventional convolutions are used for channel interactions and $k \times k \times k$ depthwise convolutions are used for local spatiotemporal interactions. As shown in Figure 4, by replacing the $3 \times 3 \times 3$ convolution with a $1 \times 1 \times 1$ traditional convolution and a $3 \times 3 \times 3$ depthwise convolution, a standard 3D bottleneck block in 3D ResNet was changed into an IP-CSN bottleneck block. This design can not only reduce parameters and FLOPs of the traditional $3 \times 3 \times 3$ convolution significantly but also preserves all channel interactions with a newly-added $1 \times 1 \times 1$ convolution.

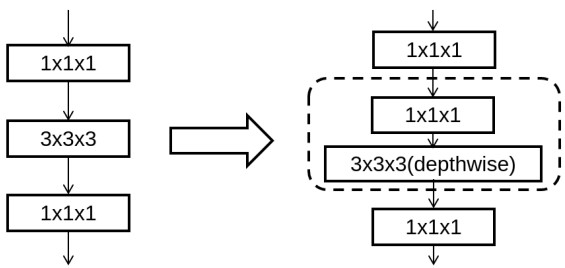

Figure 4: Example of IP-CSN bottleneck block

## Appendix C. Results for offline surgical workflow recognition

The mean accuracy, the standard deviation of the accuracy, the mean weighted Jaccard Score, and the standard deviation of the weighted Jaccard Score are calculated from video level using the test dataset and are shown in Table 4. SWNet outperforms all other methods in terms of the mean accuracy and the mean weighted Jaccard Score. The Precision, Recall, and F1-Score are shown in Table 5.

Table 4: Accuracy and Weighted Jaccard Score for offline surgical workflow recognition (mean $\pm$ std. %)

| Method | Accuracy | Weighted Jaccard Score |
|---|---|---|
| ResNetLSTM | $84.02 \pm 10.50$ | $75.97 \pm 13.27$ |
| TeCNO | $88.42 \pm 10.19$ | $81.48 \pm 12.91$ |
| EfficientNet-MSTCN | $88.31 \pm 7.778$ | $81.47 \pm 10.67$ |
| EfficientNet-MSTCN-PKNF | $88.65 \pm 8.54$ | $81.87 \pm 11.70$ |
| IPCSN-LSTM | $86.23 \pm 7.472$ | $78.21 \pm 9.84$ |
| IPCSN-LSTM-PKNF | $87.19 \pm 7.427$ | $79.36 \pm 10.32$ |
| IPCSN-MSTCN | $89.85 \pm 7.841$ | $83.47 \pm 11.12$ |
| IPCSN-MSTCN-PKNF (SWNet) | $90.60 \pm 7.288$ | $84.51 \pm 10.65$ |

Table 5: Detailed performance for offline surgical workflow recognition

| Phase Name | Method | Precision | Recall | F1-Score |
|---|---|---|---|---|
| Not a surgical phase | ResNetLSTM | 0.81 | 0.73 | 0.77 |
| | TeCNO | 0.83 | 0.82 | 0.82 |
| | EfficientNet-MSTCN-PKNF | 0.86 | 0.83 | 0.84 |
| | IPCSN-LSTM-PKNF | 0.86 | 0.78 | 0.82 |
| | IPCSN-MSTCN-PKNF(SWNet) | **0.88** | **0.85** | **0.87** |
| Ligation of short gastric vessels phase | ResNetLSTM | 0.85 | 0.89 | 0.87 |
| | TeCNO | 0.84 | 0.88 | 0.86 |
| | EfficientNet-MSTCN-PKNF | 0.90 | 0.91 | 0.90 |
| | IPCSN-LSTM-PKNF | 0.88 | **0.93** | 0.90 |
| | IPCSN-MSTCN-PKNF(SWNet) | **0.91** | 0.92 | **0.92** |
| Gastric transection phase | ResNetLSTM | 0.90 | 0.93 | 0.92 |
| | TeCNO | **0.96** | 0.94 | **0.95** |
| | EfficientNet-MSTCN-PKNF | 0.92 | **0.97** | 0.94 |
| | IPCSN-LSTM-PKNF | 0.91 | 0.96 | 0.94 |
| | IPCSN-MSTCN-PKNF(SWNet) | 0.94 | 0.96 | **0.95** |
| Bougie phase | ResNetLSTM | 0.33 | 0.40 | 0.36 |
| | TeCNO | 0.70 | 0.40 | 0.51 |
| | EfficientNet-MSTCN-PKNF | 0.61 | **0.71** | 0.65 |
| | IPCSN-LSTM-PKNF | 0.51 | 0.48 | 0.49 |
| | IPCSN-MSTCN-PKNF(SWNet) | **0.73** | 0.64 | **0.68** |
| Suturing of omentum to stomach phase | ResNetLSTM | 0.83 | 0.97 | 0.90 |
| | TeCNO | 0.86 | **1.00** | 0.92 |
| | EfficientNet-MSTCN-PKNF | **0.91** | 0.98 | 0.94 |
| | IPCSN-LSTM-PKNF | 0.87 | 0.97 | 0.92 |
| | IPCSN-MSTCN-PKNF(SWNet) | **0.91** | 0.99 | **0.95** |
| Exploration phase | ResNetLSTM | 0.09 | 0.04 | 0.06 |
| | TeCNO | 0.71 | 0.23 | 0.35 |
| | EfficientNet-MSTCN-PKNF | 0.35 | 0.18 | 0.23 |
| | IPCSN-LSTM-PKNF | 0.87 | **0.50** | **0.64** |
| | IPCSN-MSTCN-PKNF(SWNet) | **0.95** | 0.30 | 0.46 |
| Liver retraction phase | ResNetLSTM | 0.01 | 0.10 | 0.03 |
| | TeCNO | 0.42 | 0.12 | 0.19 |
| | EfficientNet-MSTCN-PKNF | 0.40 | 0.32 | 0.36 |
| | IPCSN-LSTM-PKNF | 0.66 | 0.28 | 0.40 |
| | IPCSN-MSTCN-PKNF(SWNet) | **0.81** | **0.56** | **0.66** |
| Hiatal hernia repair phase | ResNetLSTM | 0.90 | 0.65 | 0.76 |
| | TeCNO | **0.98** | 0.88 | 0.92 |
| | EfficientNet-MSTCN-PKNF | 0.93 | 0.90 | 0.92 |
| | IPCSN-LSTM-PKNF | 0.87 | 0.90 | 0.88 |
| | IPCSN-MSTCN-PKNF(SWNet) | 0.92 | **0.95** | **0.94** |
| Gastric band removal phase | ResNetLSTM | 0.81 | 0.29 | 0.43 |
| | TeCNO | 0.84 | 0.52 | 0.64 |
| | EfficientNet-MSTCN-PKNF | **0.89** | 0.41 | 0.56 |
| | IPCSN-LSTM-PKNF | 0.73 | 0.31 | 0.43 |
| | IPCSN-MSTCN-PKNF(SWNet) | **0.89** | **0.68** | **0.77** |

## Appendix D. The overview of MS-TCN

The overview of MS-TCN (Farha and Gall, 2019) is illustrated in Figure 5. Given the input $X = \{x_1, x_2, \ldots, x_t\}$, MS-TCN predicts the output $P = \{P_1, P_2, \ldots, P_t\}$ where $t$ is the current time step, $1 \leq t <= T$, $T$ is the number of total time steps, $x_t$ is the feature input at time step $t$, $P_t$ is output prediction for the current time step. In each stage of MS-TCN, $l$ is the layer number and $l \in [1, L]$, $L$ is the total number of dilated convolution layers. $S$ is the total stage number for MS-TCN. The first stage of MS-TCN only consists of temporal convolutional layers. The first layer of stage 1 is a $1 \times 1$ convolutional layer. It is used to match the input dimension with the feature map number in the network. After that, several layers of dilated 1D convolution with the same number of convolutional filters and a kernel size of 3 are used. ReLU activation is used in each layer. Residual connections are used to facilitate gradients flow. After the last dilated convolution layer, a $1 \times 1$ convolution and a softmax activation are used to get the initial predictions from the first stage. To refine the initial predictions, additional stages are used. Each additional stage takes initial predictions from the previous stage and refines them.

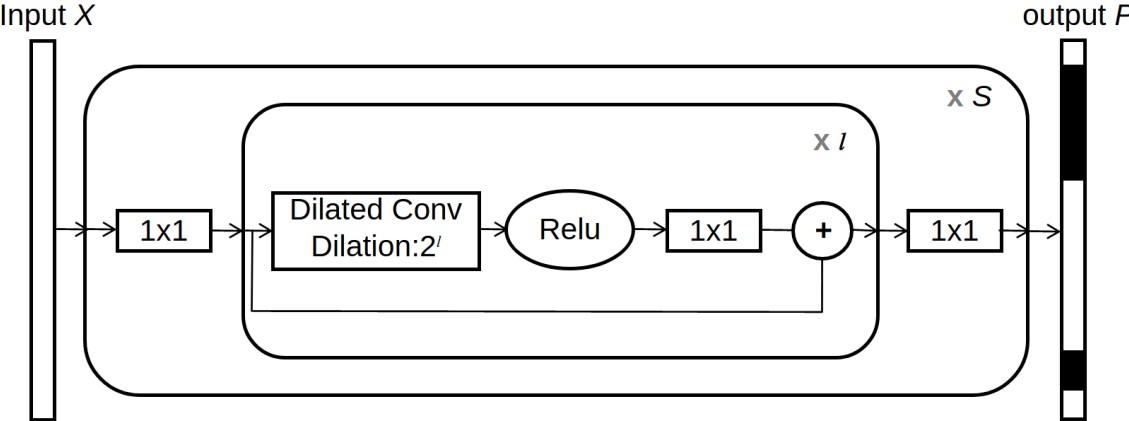

Figure 5: The overview of MS-TCN

## Appendix E. Segmental metrics

In order to evaluate out-of-order predictions and over-segmentation errors, segmental metrics (Lea et al., 2016, 2017; Farha and Gall, 2019) are utilized. We calculate the segmental edit distance score (Lea et al., 2016), and the segmental F1 score (Lea et al., 2017) at overlapping thresholds 10%, 25%, and 50%.

Let $G$ be the ground truth labeling and let $P$ be the prediction labeling. For each sequence we denote the segmental labelings $G_s$ and $P_s$ such that if $G = \{ABBBBCC\}$ where $A$, $B$, $C$ are three different labels, then $G_s = \{ABC\}$. The unnormalized segmental edit score is defined using a edit distance, $S_e(G_s, P_s)$, with insertions, deletions, and

replacements. The segmental edit score can be calculated by

$$S_{ne} = (1 - \frac{S_e(G_s, P_s)}{\max(L_G, L_P)}) \times 100 \tag{4}$$

where $L_G$ is the length for $G_s$ and $L_P$ is the length for $P_s$

For each segment, true positive and false positive are calculated by comparing its temporal Intersection over Union (IoU) with respect to the corresponding ground truth with a certain threshold. *Precision* and *Recall* are summed over all classes. The segmental F1 score can be calculated by

$$F1_s = (2 \times \frac{Precision \times Recall}{Precision + Recall}) \times 100 \tag{5}$$

