# OpenReview forum: "SWNet: Surgical Workflow Recognition with Deep Convolutional Network"
_MIDL.io/2021/Conference — MIDL 2021_

### Official Review · ~Jannis_Hagenah1 · 2021-03-04

**Confidence:** 4
**Preliminary Rating:** 2
**Recommendation:** Poster
**Final Rating:** 3

**Summary:**

In this work, the authors present a novel method for solving the problem of recognizing surgical phases from video data. Thus, they propose a three-step model, named SWNet, consisting of a feature extractor, a prediction model and a custom designed filtering method for post-processing the predictions. Furthermore, a new dataset for surgical phase recognition in sleeve gastrectomy surgery is collected and the proposed method is evaluated on this data in an online and offline setting. As a comparison, the feature extractor was replaced by a classical one which led to a drop of the model’s performance.

**Strengths:**

The paper tackles a relevant problem and even though the building blocks of the method were collected from the literature, the combination of these blocks for this application seems to be novel. The idea of integrating some kind of “causal constraints” by filtering the prediction results with a custom filter method is interesting and the evaluation results presented in this work are promising.
The collection of the large dataset is definitely worth mentioning. Following the open science spirit of MIDL and aiming on reproducibility of the presented results, I would highly recommend to make this data publicly available. The presentation is mostly good, the language is clear and the figures are helpful.


**Weaknesses:**

Even though the paper is overall sound, I have one main concern:
In my opinion, the low comparability is a serious weakness of this manuscript. Collecting a new dataset is great, and designing a new, general method is great as well. However, there are numerous published approaches to solve the given problem, and as SWNet is only evaluated on the new dataset, it is not clear how it performs compared to state-of-the-art approaches. As SWNet is presented as a general method, it should be benchmarked against competing methods! Hence, I highly recommend to apply SWNet to a publicly available dataset where performances of competing methods can be found in the literature. Thus, the authors could use data from the “EndoVis” challenge or on one of its sub-challenges (“Cataract”, “Surgical Workflow and Skill Analysis”, …). The paper would highly benefit from such a study!

Besides this main concern, I have several issues and questions:
1) I have some issues with the PKNF filtering method. Even though the motivation is clear, I did not get the details of how it works. Please revise section 2.2.3 regarding concise descriptions. How did you assess incidences or phase times? How are the thresholds applied? Please clarify.
2) Another issue related to PKNF is the surgical phase order aspect. If I understand it right, you basically apply some rule-based system in which you know what the next phase will be. This reduces the whole phase recognition problem to a detection of the frame in which this specific, already known phase starts. For this simplified problem description, the whole pipeline feels like an overkill. Could you please comment on that?
3) I did not get the method for fine-tuning. When is it performed? Why do you use completely different optimizers and hyperparameter? If the finetuning is performed end-to-end, it can be done during model training, right? Please provide more details here.
4) Did you evaluate the influence of the data augmentation methods on the model performance? I would doubt that the augmentation you proposed results in realistic variance. Specifically, horizontal or vertical flipping might lead to anatomical relationships that cannot be present in a human body or an arrangement of tools that will never happen during surgery. Hence, your model could be trained on unrealistic data and hence focus on wrong features. Could you please comment on that?
5) In 3.1, you mention the lack of training data for some classes. Did you think about performing class balancing? Would it be possible to oversample the minority classes by assessing the relevant sequences multiple times during training?
6) Could you please explain why you used different evaluation metrics for offline and online prediction? In general, the comparison of the predicted phases and the ground truth should be independent of whether the model works online or offline, right? Hence, I would expect that the metrics stay the same. Please give reason for your choice!


**Deanonymize Review:**

yes

**Detailed Comments:**

Besides the major concerns described above, I have some minor comments:
1) Regarding the data set, you state that the data comes from 14 centers and that medical experts annotated the data. It would be interesting to know more details about this, for example how much data came from which center and how many experts annotated the data. Were all annotations done by the same expert or do we expect observer-variability between different videos? An additional bonus might be one or two example frames in the appendix.
2)  In 2.2, you state that you divided the video into short segments. Did these segments overlap or were they completely separated?
3) It might be helpful to add a stronger motivation for offline prediction, as the benefit is not completely clear.
4) In 2.2.1, you state that pretrained weights for IP-CSN152 are publicly available. A reference to the resource would be very helpful for the reader!
5) Also in 2.2.1, you introduce IG-65M but neither describe nor reference it. Please add an adequate reference here.
6) In 2.2.2, Eq. 1 (the receptive field of a neuron) is nice but is does not add much information necessary to understand the method. If you need to save space, shortening this might be an option.

Additionally, there are some typos:
1) Page 2, line 7: … to filter the prediction noise…
2) Page 3, line 19: …with 9 class labels, including eight surgical…
3) Page 4, line 9: … each layer.
4) Page 4, line 20: … loss function sums the losses…
5) Page 6, line 9: …feature maps is…


**Final Rating Justification:**

The authors made great use of the rebuttal time and present a revised manuscript with a significant increase of quality. Most of all, they provide benchmark results of several state-of-the-art methods on their in-house dataset and show the promising performance of their proposed method. Furthermore, all my comments and questions were addressed and answered adequately.
Even though some parts of this paper are highly motivated by the specific commercial application of the company (PKNF, Offline Phase Detection), I think that the study is of interest to the MIDL community.
Hence, I am adjusting my rating and vote for a weak accept as well as a poster presentation at MIDL.

**Justification Of The Preliminary Rating:**

All in all, the paper presents an interesting approach with a clear motivation and promising results on the collected dataset. However, as mentioned above, I am missing the embedding into previous work as the results are not comparable to state-of-the-art methods. Hence, I cannot vote for accepting the paper in its current form. However, I want to highly encourage the authors to make good use of the rebuttal period. If you can provide an evaluation of your method on a publicly available dataset, together with a discussion of the performance in comparison to state-of-the-art-methods, the paper’s quality would rise significantly! Same holds for my major and minor concerns. I am looking forward to a revised version of this really interesting paper!

**Paper Type:**

validation/application paper

**Questions To Address In The Rebuttal:**

The most pressing point is the comparability to the state-of-the-art (also see Justification)! Hence, I would highly appreciate any further study that makes an embedding of SWNet into recent literature possible. This should be the main focus of the rebuttal.
Besides that, I listed some questions and concerns in the major and minor comments. Most of them can be solved by rephrasing and wording, so it would be great if the revised version would make these points more clear.

**Special Issue:**

no

---

> ### Author Response · Authors · 2021-03-16
> **For AnonReviewer4 (Part 1 of 2): We would like to thank the reviewer for the constructive feedback and the provided suggestions for improvement.**
>
> **Thank you so much for helping us to improve our work**. **We follow the suggestions and did a major revision**. **Here are our replies**. **Please also check our replies for all reviewers**.
>
> **This is part 1 of our replies, please also check part 2 of our replies, thank you so much**.
>
> **(1) The most pressing point is the comparability to the state-of-the-art (also see Justification)**! **Hence, I would highly appreciate any further study that makes an embedding of SWNet into recent literature possible**.
>
> Thank you so much for this great suggestion. I am so sorry about this, this is my mistake. We explain this in the “Replies for all reviewers” section (point 1). In short: we add new experiments including: TeCNO, ResNetLSTM, IPCSN-LSTM, IPCSN-LSTM-PKNF to our revision. We compare TeCNO, ResNetLSTM (recent literature) against our methods. We compare IPCSN-LSTM with IPCSN-MSTCN to show that MS-TCN is the right choice for the video action segmentation network. **With all these newly-added experiments, our methods are presented way better than our initial submission**. **Thank you so much for helping us**.
>
>
>
> **(2) I have some issues with the PKNF filtering method. Even though the motivation is clear, I did not get the details of how it works**. **Please revise section 2.2.3 regarding concise descriptions**. How did you assess incidences or phase times? How are the thresholds applied? Please clarify.
>
> Thank you for this great advice. We revised section 2.2.3 with more details. We first calculate the minimum phase time T for each surgical phase from our ground truth annotation (training data). The connection threshold is set according to the minimum phase time T. In this work, we set the connection threshold to be 40% of the minimum phase time T. For phase incidence, we also calculate the minimum phase incidence for each surgical phase from our ground truth annotation (training data).
>
>
>
> **(3) Another issue related to PKNF is the surgical phase order aspect**. **If I understand it right, you basically apply some rule-based system in which you know what the next phase will be**. **This reduces the whole phase recognition problem to a detection of the frame in which this specific, already known phase starts**. **For this simplified problem description, the whole pipeline feels like overkill**. **Could you please comment on that**?
>
> This is a great question. When we check on our dataset, we find out several surgical phases do not follow step order at all. Currently, we only focus on the steps that follow step order for example “Exploration phase”. We know the “Exploration phase” can only happen at the beginning of the case for sure. It is clear that predictions of the “Exploration phase” in the middle or at the end of the case are wrong predictions and need correction.
>
>
>
> **(4) I did not get the method for fine-tuning**. **When is it performed**? **Why do you use completely different optimizers and hyperparameter**? If the finetuning is performed end-to-end, it can be done during model training, right? Please provide more details here.
>
> All our experiments are 2-stage training, we can train each part of the model separately with different optimizers and different hyper-parameters. We first conduct transfer learning for CNN/3DCNN on our video dataset. We then extract features with CNN/3DCNN. Now we have a feature dataset. We use those extracted features to train our LSTM/MSTCN network separately. Training different parts of the pipeline, we utilize different optimizers and hyper-parameters. For example, SGD optimizer is used for training IPCSN, Adam optimizer is used for training MS-TCN.
>
>
>
> **(5) Did you evaluate the influence of the data augmentation methods on the model performance**? **I would doubt that the augmentation you proposed results in realistic variance**. **Specifically, horizontal or vertical flipping might lead to anatomical relationships that cannot be present in a human body or an arrangement of tools that will never happen during surgery**. **Hence, your model could be trained on unrealistic data and hence focus on wrong features**. **Could you please comment on that**?
>
> This is a great question. Although I did not evaluate this with IP-CSN, I did previous play around with training I3D with the same dataset for the same surgical workflow recognition problem. I3D with data augmentation (rotation, flipping, sampling fps change) performs better than I3D without data augmentation by less than 1%. Data augmentation like affine and shear will decrease accuracy. I kind of forget how much decrease it costs from the accuracy aspect. I definitely should investigate this data augmentation problem in the future. Here is a nice paper I think will help me to keep exploring this field:
>
> Lin, Zhiqiu, et al. "Visual chirality." Proceedings of the IEEE/CVF Conference on Computer Vision and Pattern Recognition. 2020.

---

> > ### Author Response · Authors · 2021-03-16
> > **For AnonReviewer4 (Part 2 of 2): We would like to thank the reviewer for the constructive feedback and the provided suggestions for improvement.**
> >
> > **Thank you so much for helping us to improve our work**. **We follow the suggestions and did a major revision**. **Here are our replies**. **Please also check our replies for all reviewers**.
> >
> > **This is part 2 of our replies, please also check part 1 of our replies, thank you so much**.
> >
> > **(6) In 3.1, you mention the lack of training data for some classes**. **Did you think about performing class balancing**? **Would it be possible to oversample the minority classes by assessing the relevant sequences multiple times during training**?
> >
> > This is another great question. Although I did not evaluate this with IP-CSN, I did try SMOTE with I3D before, the accuracy decrease by around 3%. Direct predictions from I3D are very noisy. For this work, we kind of sample according to annotation segments. We sample 5 video segments for each annotation segment in each video during each epoch of training. Because each segment looks different from each other, we kind of balancing our data there.
> >
> > **(7) Could you please explain why you used different evaluation metrics for offline and online prediction**?
> >
> > I am so sorry about this, this is my mistake. We explain this in the “Replies for all reviewers” section(point 2). In short: The performance between different methods is similar from the overall accuracy and the weighted Jaccard Score aspects. In order to evaluate out-of-order predictions and over-segmentation errors, segmental metrics are utilized. We add this part to our new revision and add some related references.
> >
> > **(8) Regarding the data set, you state that the data comes from 14 centers and that medical experts annotated the data**. **It would be interesting to know more details about this**.
> >
> > Each video in our dataset is assigned to one main annotator. Two other annotators check the annotation after the video is annotated by the main annotator. Surgeons perform differently in this dataset. The step order and step time can be different. In this work, from the data design aspect, our data contains a “Not a surgical phase” label. This is quite different from other datasets. We annotate each surgical phase as accurately as possible. We do not merge our surgical phases with other unrelated surgical phases, long-time idle, or other unrelated surgical activity. I add more details about the “Not a surgical phase” label in our revision. More medical experts are working on the dataset now. More medical-research-based annotation will be added.
> >
> > **(9) In 2.2, you state that you divided the video into short segments**. **Did these segments overlap or were they completely separated**?
> >
> > The segments overlap with each other. Our prediction interval is one second. Every second we grab a 19.2 s video segment for IP-CSN to process.
> >
> > **(10) It might be helpful to add a stronger motivation for offline prediction, as the benefit is not completely clear**.
> >
> > The reason we focus on offline prediction is that this is one of our projects at our company. Our company provides feedback to surgeons. Surgeons upload videos to us, we segment out the videos for further machine learning analyses and for the expert review process. We try to help surgeons to improve the skills. The offline surgical workflow recognition prediction results are usually better than online surgical workflow recognition prediction results, it can help us benchmark surgeons more accurately. This work is an industrial application. That is the reason why we focus on PKNF and smoothing loss (over-segmenting can be a problem for our users), we want to make sure the models can provide a good user experience when it is deployed in production.
> >
> > **(11) In 2.2.1, you state that pretrained weights for IP-CSN152 are publicly available**. **A reference to the resource would be very helpful for the reader**!
> >
> > This is a great suggestion, I add the link in the reference in the new revision.
> >
> > **(12) Also in 2.2.1, you introduce IG-65M but neither describe nor reference it**. **Please add an adequate reference here**.
> >
> > Thank you so much for this advice, I add a reference for IG-65M in the new revision. IG-65M is a dataset has 65 millions of video, IP-CSN needed a large amount of data to train. All these deep 3DCNNs need a huge dataset to train. Accuracy improves a lot just by changing the number of training videos from 150 to 300 (Previous experience with I3D, of course, this might also be related to the difficulty level of the problem).
> >
> > **(13) In 2.2.2, Eq. 1 (the receptive field of a neuron) is nice but it does not add much information necessary to understand the method**. **If you need to save space, shortening this might be an option**.
> >
> > Thank you so much for this great advice. I have deleted this equation and I also send part of the MS-TCN section to the appendix to save some space for the newly added experiments.
> >
> > **(14) Typos**
> >
> > Thank you so much for helping us. I have fixed the typos and invite a native English speaker to help me with the paper.

---

> > > ### Comment · AnonReviewer4 · 2021-03-22
> > > **Follow-Up Questions**
> > >
> > > Thank you very much for the clarifications and the detailed rebuttal. I just have a short follow-up question on comment (9), regarding the overlapping video sequences:
> > >
> > > As you divided your dataset into training and test data on the full video level, it is not possible that two sequences with huge overlap will be split up so that one of them is in the training and one in the test dataset, right?
> > >
> > > And regarding the comparably long sequence length, there will be ambiguous sequences when the surgical phase changes. How did you care for the time slot right before the next surgical phase starts? Did you cut them out? And if yes, do you expect systematical errors by always ignoring the last 19 seconds of a surgical phase that might carry relevant information (for example removing a tool)?

---

> > > > ### Author Response · Authors · 2021-03-22
> > > > **Thank you so much for helping us to improve our work.**
> > > >
> > > > Here are our replies to the follow-up questions. Thanks again for helping us.
> > > >
> > > > **(1) As you divided your dataset into training and test data on the full video level, it is not possible that two sequences with huge overlap will be split up so that one of them is in the training and one in the test dataset, right?**
> > > >
> > > > We have a total of 461 videos. We split it into 3 datasets. 317 videos for training (training set), 82 videos for validation (validation set), and 62 videos for testing (test set).
> > > >
> > > > During training, we only use samples from the training set (The dataset that has 317 videos). During validation, we only use samples from the validation set (The dataset that has 82 videos). During testing, we only use full videos from the test set to test (The dataset that has 62 videos).
> > > >
> > > > In this way, the test dataset is completely new to all of our models.
> > > >
> > > > **(2) And regarding the comparably long sequence length, there will be ambiguous sequences when the surgical phase changes. How did you care for the time slot right before the next surgical phase starts? Did you cut them out? And if yes, do you expect systematical errors by always ignoring the last 19 seconds of a surgical phase that might carry relevant information (for example removing a tool)?**
> > > >
> > > > This is a great question. In short: MS-TCN decides when the phase starts and when the phase ends.
> > > >
> > > > In detail: A 19.2 seconds window seems to be a long window. At first glance, I was worried about it can cause a problem with the start and end time for surgical phases.  But because every one second, we grab a 19.2 seconds window segment for IP-CSN to process (The video segments overlap). We have a feature from IP-CSN to represent what is happening every one second of the video. We use full video features to train MS-TCN. MS-TCN will help to make the decision on when is the phase start and when is the phase end. This is similar to the MS-TCN paper [1] (The authors also use I3D to extract features from short video clips in a similar way).
> > > >
> > > > Please also check on Figure 2. In video 1 and video 2 (see the start and end time of the orange phase), it is clear that the features from 3DCNN (IP-CSN) can help the MS-TCN identify phase transactions better.  Table 5 in Appendix C also shows SWNet (IPCSN trained with 19.2s window) performs better in most of the surgical phases than other methods.
> > > >
> > > > Why did I choose this 19.2s window? I ask our medical experts: What is the video clip length needed for surgeons to identify the surgical phase from a single video clip? The answers I got are around 15s to 20s. (There are very simple phases, one frame is good enough, but the answer is for all surgical phases in most common procedures). So I select the 19.2s window.
> > > >
> > > > But I agree that a very long window can also cause problems to very short surgical action (I personally do not think a 19.2s window can work well for surgical action that only lasts 5 seconds. I did not test about this.).  I will conduct a deep error analysis in the future.
> > > >
> > > > There is a lot of future work I can explore for utilizing the 3DCNN. (1) 3DCNN needs a huge dataset to train. I will investigate different training methods to make sure 3DCNN can also work well for small datasets. (2) 3DCNN trains with long windows might not suitable for identifying surgical actions that are very fast. The best window length is something I need to work on in the future.
> > > >
> > > > Reference:
> > > > [1] Farha, Yazan Abu, and Jurgen Gall. "Ms-tcn: Multi-stage temporal convolutional network for action segmentation." Proceedings of the IEEE/CVF Conference on Computer Vision and Pattern Recognition. 2019.

---

> > > > > ### Comment · AnonReviewer4 · 2021-03-22
> > > > > **Thanks for fast Clarification**
> > > > >
> > > > > Thank you for the fast clarification and answering my questions. This really helped.

---

### Official Review · AnonReviewer3 · 2021-03-08

**Confidence:** 3
**Preliminary Rating:** 2
**Final Rating:** 3

**Summary:**

The authors proposed a SWNet to address the video segmentation problem for surgical workflow. They used IP-CSN for feature extraction and MS-TCN for video segmentation. They also included a post-processing step for surgical workflow called Prior Knowledge Noise Filtering (PKNF) to further modify the output predictions. The experiments showed their network can generate good performance in the task.

**Strengths:**

1. The authors are familiar with the state-of-the-art methods in video-related action recognition and feature extraction methods, which are suitable in the surgical workflow segmentation problem.
2. The post-processing step (aka Prior Knowledge Noise Filtering) includes a special design of a filter that can filter out the unexpected labels considering the surgical workflow.

**Weaknesses:**

1. The work seems to be a combination of previous methods in action recognition plus post-processing. The design of PKNF is helpful but the novelty is marginal.
2. The authors need to be more clear about the metrics of the problem.
3. The ablation study seems to focus on comparing the 3D convnet and 2D efficient net, which may not be able to prove the effectiveness of the present model.
4. The proposed network is not compared with more recent network components nor in a public dataset, which makes the result hard to evaluate or repeat.

**Deanonymize Review:**

no

**Final Rating Justification:**

Thank you for the update, the new experiments brought the paper to a much solid level and I would like to accept this paper.

**Justification Of The Preliminary Rating:**

1. The work seems to be a combination of previous methods in action recognition plus post-processing. The design of PKNF is helpful but the novelty is marginal.
2. The experiments are a little bit biased to test 3D networks vs 2D networks, and the video segmentation network was not compared with other works.

**Paper Type:**

validation/application paper

**Questions To Address In The Rebuttal:**

1. More ablation studies in more related works, including the MS-TCN part.
2. More novel designs related to the surgical workflow segmentation problem.

**Special Issue:**

no

---

> ### Author Response · Authors · 2021-03-16
> **For AnonReviewer3:   We would like to thank the reviewer for the encouraging and constructive comments. We highly appreciate the feedback.**
>
> **Thank you so much for helping us to improve our work**. **We did a major revision on our work**. **Here are our replies**. **Please also check our replies for all reviewers**.
>
>
> **(1) The design of PKNF is helpful but the novelty is marginal**.
>
> We explain this in the “Replies for all reviewers” section (point 4). In short: PKNF is necessary for our industrial application. We add more experiments in Table 1, PKNF can provide around 0.5% to 1.5% improvement from accuracy.
>
>
> **(2) The authors need to be more clear about the metrics of the problem**.
>
> This is completely my mistake. I am so sorry about this. We explain this in the “Replies for all reviewers” section (point 2). In short: The performance between different methods is similar from the overall accuracy and the weighted Jaccard Score aspects in online surgical workflow recognition. In order to evaluate out-of-order predictions and over-segmentation errors, segmental metrics are utilized. We add more details and references to our new revision. Thank you so much for this great suggestion.
>
>
>
> **(3) The ablation study seems to focus on comparing the 3D convnet and 2D efficient net, which may not be able to prove the effectiveness of the present model**. (More novel designs related to the surgical workflow segmentation problem.)
>
> I failed to communicate clearly about this in the paper. I have explained this in the “Replies for all reviewers” section (point 3). In short: For feature extraction backbone selection, we want to use SOTA against SOTA (3DCNN STOA vs 2DCNN SOTA, we know 3DCNN SOTA might be a better choice, but we also want to know where does 3DCNN perform better.) For the innovation design part, we utilize SOTA to build up our pipeline instead of applying existed methods to solve surgical workflow recognition problems. Our data is annotated differently compares to other surgical workflow recognition problems. We believe this can lead to discussions about computer-vision-based annotation and medical-research-based annotation.
>
>
> **(4) The proposed network is not compared with more recent network components nor in a public dataset, which makes the result hard to evaluate or repeat**. (More ablation studies in more related works, including the MS-TCN part.)
>
> I am so sorry about this, this is my mistake. We explain this in the “Replies for all reviewers” section(point 1). In short: we add new experiments including: TeCNO, ResNetLSTM, IPCSN-LSTM, IPCSN-LSTM-PKNF to our revision. We compare TeCNO, ResNetLSTM with our methods. We compare IPCSN-LSTM with IPCSN- MS-TCN to show that MS-TCN is the right choice for the video action segmentation network. **These newly-added results help us present our ideas and designs in a much better way**. **Thank you so much**.

---

### Official Review · AnonReviewer2 · 2021-03-09

**Confidence:** 3
**Preliminary Rating:** 2
**Final Rating:** 3

**Summary:**

The authors present a Deep Learning Framework named SWNet for spatio-temporal tracking from surgical videos for workflow recognition. Their pipeline utilizes Interaction-Preserved Channel-Separated Convolutional Network (IP-CSN) for feature extraction in video segments. This is followed by a Multi-Stage Temporal Convolutional Network (MS-TCN) that aggregate the temporal features from individual segments. This is followed by a Prior Knowledge Noise Filtering (PKNF) module to suppress noise from the predictions during offline prediction.  They also provide a modified workflow for the online prediction setup, using a smoothing loss function in conjunction with weighted cross entropy. They demonstrate that this approach provides state-of-the-art performance, as measured by the accuracy and the weighted Jaccard Index.

**Strengths:**

The paper is laid out clearly with the main ideas explained precisely. The authors have also provided supporting ablation studies which illustrate the utility of each individual module they adopt. Their model performs better than the baseline in the offline surgical workflow recognition task. The adaptation using L2 smoothing provides reasonable performance improvements in the online setting as well.

**Weaknesses:**

1. The demonstrated gains over their efficient net-PKNF baseline, and their own method without PKNF in terms of Table 1 are very marginal, with differences being in the second decimal for precision , recall and F1. Currently, the results as presented do not make a strong enough case for using PKNF, especially since the authors do not provide standard deviations for their accuracies. For example

2. The authors have randomly split their dataset into train, test and validation, which may lead to biased results due to sampling.

3. The authors do not provide enough explanation for the evaluation metrics used, especially in the online setup. This makes it hard to gauge the magnitude of the actual improvements obtained

**Deanonymize Review:**

no

**Detailed Comments:**

The main weakness of the paper is in the presentation of the results section

Suggestions:

1. Performing k-fold cross validation as opposed to a random splits for evaluations.
2. Highlighting improved performance clearly in Table 2.
3. Extending Table 2 for all comparisons in Table 1
4. Explaining the metrics used for evaluation in the online prediction
5. Table 3 does not compare against any baseline, which makes the gains hard to put into context.

**Final Rating Justification:**

Thanks for the clarifications, both during the rebuttal and with the follow-up. I'm willing to upgrade my initial score. However, I'd like the authors to consider adding details to the paper for all the points in the review where more intuition/explanation was requested.

**Justification Of The Preliminary Rating:**

The proposed framework mainly combines ideas from existing literature, rather than proposing a novel methodology. The main reason for the provided rating is the lack of clear and conclusive results as presented, especially given that the paper is a validation/application submission. Additionally, the results seem to not be adequately compared with existing literature studying this application.

**Paper Type:**

validation/application paper

**Questions To Address In The Rebuttal:**

1. Could the authors please provide standard deviation measures and statistical metrics for improved performances over baselines and ablation studies.
2. Why has the efficient net architecture been chosen as a baseline? The comparison for the same in terms of overall accuracy with and without PKNF currently shows differences in the 3rd decimal, which is not convincing.
3. How is \lambda chosen for the online setting?
4. How many parameters does the proposed SWNet Model have in comparison with EfficientNet?

**Special Issue:**

no

---

> ### Author Response · Authors · 2021-03-16
> **For AnonReviewer2 :   We would like to thank the reviewer for the great comments and insightful thoughts.**
>
> **Thank you so much for helping us to improve our work**. **Here are our replies**. **Please also check our replies for all reviewers**. **Thanks again**.
>
> **(1) The demonstrated gains over their efficientnet-PKNF baseline and their own method without PKNF in terms of Table 1 are very marginal. The authors do not provide standard deviations for their accuracies**. (Could the authors please provide standard deviation measures and statistical metrics for improved performances over baselines and ablation studies.)
>
> We explain this in the “Replies for all reviewers” section (point 4). In short: PKNF is necessary for our industrial application. We add more experiments in Table 1, PKNF can provide around 0.5% to 1.5% improvement from the accuracy aspect.
>
> Thank you for this great suggestion.  Mean accuracies and the standard deviations for the accuracies are calculated in Table 4 in the revision. Due to lack of space, we currently arrange it in Appendix C.
>
>
>
> **(2) The authors have randomly split their dataset into train, test, and validation, which may lead to biased results due to sampling**.
>
> We are so sorry about this part. Training feature extraction backbone (3DCNN) takes more than 2 weeks. Extract features will take several days. Train action segmentation model will take 1 day.  We do not have enough time for doing K-folder cross-validation during the rebuttal. But we are able to provide Standard deviations for the accuracies in Table 4 in the revision as you suggested. Thank you so much for providing great suggestions to help us improve our work.
>
>
>
> **(3) The authors do not provide enough explanation for the evaluation metrics used, especially in the online setup**. **This makes it hard to gauge the magnitude of the actual improvements obtained**.
>
> I am so sorry about this, this is my mistake. We explain this in the “Replies for all reviewers” section (point 2). In short: The performance between different methods is similar from the overall accuracy and the weighted Jaccard Score aspects. In order to evaluate out-of-order predictions and over-segmentation errors, segmental metrics are utilized. We add this part to our new revision. Thank you so much for this great suggestion.
>
>
>
> **(4) Highlighting improved performance clearly in Table 2**. **Extending Table 2 for all comparisons in Table 1**.
>
> Thank you for these great suggestions. We highlight performance clearly in our new revision. Due to lack of place, we currently arrange an extended version of Table 2 in Appendix C and renamed it Table 5. **These newly-added results help us present our methods in a much better way**. **Thank you so much**.
>
>
>
> **(5) Table 3 does not compare against any baseline, which makes the gains hard to put into context**. (**the results seem to not be adequately compared with existing literature studying this application**)
>
> I am so sorry about this, this is my mistake. We explain this in the “Replies for all reviewers” section (point 1). In short: we add new experiments including: TeCNO, ResNetLSTM, IPCSN-LSTM, IPCSN-LSTM-PKNF to our revision. Thank you so much for this great suggestion.
>
>
>
> **(6) Why has the efficient net architecture been chosen as a baseline**?
>
> I failed to communicate this correctly in our paper. We explain this in the “Replies for all reviewers” section (point 3). In short: For feature extraction backbone selection, we want to use SOTA against SOTA (3DCNN STOA vs 2DCNN SOTA, we know 3DCNN SOTA might be a better choice, but we also want to know where does 3DCNN perform better.)
>
>
>
> **(7) How is \lambda chosen for the online setting**?
>
> We start by using 0.15 from MS-TCN paper. We calculate segmental metrics to evaluate the smoothing effect to determine if we increase or decrease this value.
>
>
>
> **(8) How many parameters does the proposed SWNet Model have in comparison with EfficientNet**?
>
> This is a great question. The two methods we use are IPCSN-MSTCN-PKF(SWNet) and EfficientNet-MSTCN-PKNF. IP-CSN152 was used, it has 32M parameters in SWNet. EfficientNetB5 is used in EfficientNet-MSTCN-PKNF, it has 30M parameters. The MS-TCN part is the same for both methods.

---

### Author Response · Authors · 2021-03-16
**Replies for all reviewers (Part 1 of 3) We want to thank the reviewers for providing many valuable suggestions on our paper.**

**We want to thank the reviewers for providing many valuable suggestions on our paper**. **We did a major revision for our paper**. **We followed the suggestions from the reviewers and update our work**. **Please allow me to address several important comments here**.

**This is part 1 of our replies for all reviewers, please also check part 2 and part 3, thank you so much**.

**(1) Did not compare with existing literature studying**. (The most pressing point)

This is definitely my mistake. I definitely should conduct more experiments.

In the updated revision, for offline surgical workflow recognition, we compare our design with two existing literature studying: ResNetLSTM [1] (The most common method) and TeCNO [2] (2020 SOTA). The EfficientNet-MSTCN is supposed to be an upgraded version of TeCNO for this task, but I failed to communicate that clearly and I did not prove that in our old version. Now, with the newly added experiments, we can compare our methods with the existing literature studying. For online surgical workflow recognition, we compare our design with ResNetLSTM and TeCNO.

I also take using MS-TCN [3] as granted for video action segmentation. I also need to prove that MS-TCN is a better choice, so I replace MS-TCN with LSTM in SWNet and compare the results in the updated revision. (Please check Section 3 as well as Appendix C. in our revision)

**(2) Did not explain the metrics used for evaluation in the online prediction**. (Why it is different from the offline prediction)?

This is another mistake I made. I should explain clearly to the reader. I failed to communicate clearly on this. For online surgical workflow recognition, the performance for TeCNO and IPCSN-MSTCN are similar from the overall accuracy and the weighted Jaccard Score aspects. Models achieving similar accuracy may have large differences, as visualized in Figure 3 in our revision. Frame-wise metrics are not suitable to evaluate over-segmentation errors. In order to evaluate out-of-order predictions and over-segmentation errors, segmental metrics [3,4,5] are utilized. We select those metrics because they are used in video action segmentation. We can utilize them to evaluate the smoothing effect just like the MS-TCN paper. I added some explanations and references in our revision. (Please check Section 3.2 in our revision)

---

> ### Author Response · Authors · 2021-03-16
> **Replies for all reviewers (Part 2 of 3) We want to thank the reviewers for providing many valuable suggestions on our paper.**
>
> **This is part 2 of our replies for all reviewers, please also check part 1 and part 3, thank you so much**.
>
> **(3) Seems to focus on comparing the 3D convnet and 2D efficient net**. (Why efficient net?) **More novel design**.
>
> Yes, the paper did focus on comparing the feature extraction backbone (IPCSN vs efficient net). Please allow me to explain our design ideas in detail. We believe that the surgical workflow recognition problem is both an action recognition problem and video action segmentation problem. Our paper is an industrial application paper. We begin the design of our workflow with a literature review.
>
> For action recognition, we find out on the K400 dataset [6]:
>
> IPCSN > R(2+1)D > SlowFast > I3D-NL > Two stream I3D ([7])
>
> For action segmentation, we find out on 50 salads / GTEA:
>
> MS-TCN > ED-TCN > Bi-LSTM ([3])
>
> For image classification:
>
> EfficientNet-B5 > InceptionResNetV2 > ResNet ([8])
>
> We want to provide the best to our customers, so we want to apply the SOTA in each category. In the original MS-TCN work, the author utilizes Two stream I3D + MSTCN, because the optical flow stream in I3D [10] needs expensive computation. Replacing Two stream I3D with IP-CSN seems to be a good choice for us. In theory, we believe that IPCSN+MSTCN should perform better. EfficientNet+MSTCN should perform better than ResNet+MSTCN (TeCNO).
>
> Training 3DCNN needs a lot of data. IP-CSN is trained with IG65M [9] (65 million videos) to become the SOTA. One reviewer mentioned about “EndoVis” challenge. If I remember correctly, SlowFast is utilized in one of the challenges in 2020. I3D is utilized in one of the challenges in 2019. But they failed to perform better than the CNN-RNN design. This seems to be contradicted to the I3D paper [10]. The reason might be: 3DCNN needs a huge dataset to train. So, we collect a large dataset in order to train our SWNet. Now we can conduct experiments to use SOTA against SOTA (3DCNN STOA vs 2DCNN SOTA, we know 3DCNN SOTA might be a better choice, but we also want to know where does 3DCNN perform better.) That is the reason why part of the paper is comparing the 3D convnet and 2D efficient net. We apply SOTA in each block of our pipeline, improving the pipeline is kind of difficult. We are working on and we will keep working on improving the pipeline.
>
> In this work, from the method design aspect, we try to use 3DCNN and train on a large dataset. Instead of applying the original I3D-MSTCN [3], ResNet-MSTCN (TeCNO) [2], or ResNetLSTM [1] to our problem, we build IPCSN-MSTCN and EfficientNet-MSTCN. In order to filter the noise, we design a simple filtering algorithm named PKNF. We investigate the difference between utilizing 2DCNN SOTA (EfficientNet) as the feature extraction backbone and utilizing 3DCNN SOTA (IP-CSN) as the feature extraction backbone. Our work proves SOTA 3DCNN can also perform well when it is trained correctly, this can lead to a discussion about adopting SOTA 3DCNN to this field.
>
> In this work, from the data design aspect, our data contains a “Not a surgical phase” label. This is also very different from other works in this field. We annotate each surgical phase as accurately as possible. We do not merge our surgical phase with other unrelated surgical phases, long-time idle, or other unrelated surgical activity. “Not a surgical phase” includes a lot of other surgical phases we do not annotate or lack of data to train, for example: “Adhesysios”, “Dissection of angle of his”, “Sparing of pylorus”, “Specimen retrieval” and so on. “Not a surgical phase” also includes phase transaction time. “Not a surgical phase” also includes surgical activities like long-time idle, long-time out-of-body, surgeon inspecting the organ, and so on. We believe this can lead to a discussion about computer-vision-based annotation and medical-research-based annotation.

---

> > ### Author Response · Authors · 2021-03-16
> > **Replies for all reviewers (Part 3 of 3) We want to thank the reviewers for providing many valuable suggestions on our paper.**
> >
> > **This is part 3 of our replies for all reviewers, please also check part 1 and part 2, thank you so much**.
> >
> > **(4) The comparison for the same in terms of overall accuracy with and without PKNF currently shows differences in the 3rd decimal, which is not convincing**. (The design of PKNF is helpful but the novelty is marginal. )
> >
> >
> > I failed to communicate clearly that our work focuses on industrial application. Thank you for pointing this out. Please allow me to explain why PKNF is important for our industrial application.
> >
> > We cannot send noisy machine learning predictions to our customers. Although small noise does not affect accuracy a lot, it really affects our user experiences.
> >
> > From the improvement aspect,
> >
> > EfficientNet-MSTCN-PKNF outperforms EfficientNet-MSTCN by 0.43% in overall accuracy.
> >
> > IPCSN-LSTM-PKNF outperforms IPCSN-LSTM by 1.65% in overall accuracy.
> >
> > IPCSN-MSTCN-PKNF(SWNet) outperforms IPCSN-MSTCN by 1.16% in overall accuracy.
> >
> > Because our full pipeline utilizes SOTA design in each block, around 0.5% to 1.5% improvement in the pipeline by PKNF is not very bad.
> >
> > From the PKNF design aspect, we talk about three design ideas from medical aspects. We believe this can lead to a good discussion.
> >
> >
> >
> > Our work is filled with industry design ideas. Although our main focus is offline surgical workflow recognition, we investigate online surgical workflow recognition as well. We focus on providing smoothed predictions. We want to improve customers' experience. It is also the reason why we built the pipeline with SOTA designs. I failed to communicate clearly about our design in our initial submission. This is my mistake.
> >
> >  **We would like to thank all reviewers again for providing great insights to help us present our ideas in a much better way**. **I hope this major revision can meet the team’s expectations**.
> >
> >
> >
> >
> > Reference:
> >
> > [1] Jin, Yueming, et al. "SV-RCNet: workflow recognition from surgical videos using recurrent convolutional network." IEEE transactions on medical imaging 37.5 (2017): 1114-1126.
> >
> > [2] Czempiel, Tobias, et al. "TeCNO: Surgical Phase Recognition with Multi-Stage Temporal Convolutional Networks." International Conference on Medical Image Computing and Computer-Assisted Intervention. Springer, Cham, 2020.
> >
> > [3] Farha, Yazan Abu, and Jurgen Gall. "Ms-tcn: Multi-stage temporal convolutional network for action segmentation." Proceedings of the IEEE/CVF Conference on Computer Vision and Pattern Recognition. 2019.
> >
> > [4] Li, Shi-Jie, et al. "Ms-tcn++: Multi-stage temporal convolutional network for action segmentation." IEEE Transactions on Pattern Analysis and Machine Intelligence (2020).
> >
> > [5] Lea, Colin, et al. "Temporal convolutional networks for action segmentation and detection." proceedings of the IEEE Conference on Computer Vision and Pattern Recognition. 2017.
> >
> > [6] Kay, Will, et al. "The kinetics human action video dataset." arXiv preprint arXiv:1705.06950 (2017).
> >
> > [7] Tran, Du, et al. "Video classification with channel-separated convolutional networks." Proceedings of the IEEE/CVF International Conference on Computer Vision. 2019.
> >
> > [8] Tan, Mingxing, and Quoc Le. "Efficientnet: Rethinking model scaling for convolutional neural networks." International Conference on Machine Learning. PMLR, 2019.
> >
> > [9] Ghadiyaram, Deepti, Du Tran, and Dhruv Mahajan. "Large-scale weakly-supervised pre-training for video action recognition." Proceedings of the IEEE/CVF Conference on Computer Vision and Pattern Recognition. 2019.
> >
> > [10] Carreira, Joao, and Andrew Zisserman. "Quo vadis, action recognition? a new model and the kinetics dataset." proceedings of the IEEE Conference on Computer Vision and Pattern Recognition. 2017.

---

### Meta-Review · Area_Chair1 · 2021-03-29

**Recommendation:** Accept (Oral)

**Metareview:**

The reviewers agree that the paper provides an in-depth evaluation of an interesting problem. Albeit with little methodological novelty, this is one of the strongest validation papers in my pile. The reviewers have all upgraded their rating after the rebuttal phase, thereby highlighting the strong responses provided to the initial queries.

**Paper Type:**

validation/application paper

---

### Decision · Program_Chairs · 2021-03-31

Accept